# "Train one, Classify one, Teach one" - Cross-surgery transfer learning for surgical step recognition

**Daniel Neimark**[1]                                                    DANIELN@THEATOR.IO
**Omri Bar**[1]                                                          OMRI@THEATOR.IO
**Maya Zohar**[1]                                                        MAYA@THEATOR.IO
**Gregory D. Hager**[2,1]                                                HAGER@CS.JHU.EDU
**Dotan Asselmann**[1]                                                   DOTAN@THEATOR.IO

[1] *Theator Inc., Palo Alto, CA, USA.*

[2] *Department of Computer Science, Johns Hopkins University, Baltimore, USA.*

## Abstract

Prior work demonstrated the ability of machine learning to automatically recognize surgical workflow steps from videos. However, these studies focused on only a single type of procedure. In this work, we analyze, for the first time, surgical step recognition on four different laparoscopic surgeries: Cholecystectomy, Right Hemicolectomy, Sleeve Gastrectomy, and Appendectomy. Inspired by the traditional apprenticeship model, in which surgical training is based on the *Halstedian* method, we paraphrase the "*see one, do one, teach one*" approach for the surgical intelligence domain as "*train one, classify one, teach one*". In machine learning, this approach is often referred to as transfer learning. To analyze the impact of transfer learning across different laparoscopic procedures, we explore various time-series architectures and examine their performance on each target domain. We propose a Time-Series Adaptation Network (TSAN), an architecture optimized for transfer learning of surgical step recognition. In addition, we show how TSAN can be pre-trained using self-supervised learning on a Sequence Sorting task. Such pre-training enables TSAN to learn workflow steps of a new laparoscopic procedure type given only a small number of samples from the target procedure dataset. Our proposed architecture leads to better performance compared to other possible architectures, reaching over 90% accuracy when transferring from laparoscopic Cholecystectomy to the other three procedure types.

**Keywords:** Surgical Intelligence, Surgical Transfer Learning, Surgical Step Recognition, Phase Recognition, Domain Adaptation, Deep Learning.

## 1. Introduction

Minimally invasive surgery (MIS) video analysis is steadily gaining acceptance for surgical competency assessment (Ritter et al., 2019; Feldman et al., 2020). As MIS is performed under visualization of endoscopic footage, the possibilities for AI-enabled computer-assisted surgery (CAS) applications are immense (Maier-Hein et al., 2017).

A variety of surgery-related video-analysis tasks have been explored in recent studies. Surgical step (phase) recognition (Bar et al., 2020; Twinanda et al., 2016; Zisimopoulos et al., 2018; Hashimoto et al., 2019), surgical tool detection and segmentation (Twinanda et al., 2016; Al Hajj et al., 2019; Choi et al., 2017; Ni et al., 2020; Jin et al., 2018) and surgical

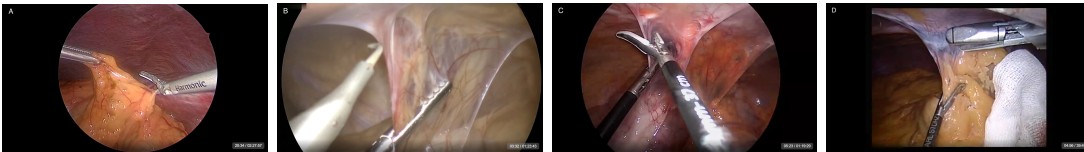

Figure 1: The same step, *Adhesiolysis*, is viewed in different procedures. (**A**) Cholecystectomy, (**B**) Appendectomy, (**C**) Right Hemicolectomy, and (**D**) Sleeve Gastrectomy.

gesture and skill assessment (Gao et al., 2014; Ahmidi et al., 2017) are a few examples. However, these studies were developed and evaluated on a single type of procedure.

Regardless of the use case there is value in creating video-based applications that are able to readily scale to serve a wide variety of surgical procedures. This will both broaden the impact of AI-based systems and also create more relevant, actionable tools that meet the needs of the broadest population of surgeons.

This study aims to address three key aspects which, taken together, provide insight into our practical ability to scale video analysis of surgery to multiple procedures while minimizing the need for large labeled datasets. First, we assess the potential of using self-supervised pre-training to reduce dependence on explicitly labeled data. Second, we investigate the effectiveness of transfer learning to move pre-trained models between different surgical procedures. Finally, we explore the impact of data size on the adaptation capabilities. Taken together, our results suggest a practical and effective path to generalizing video analysis of surgery while minimizing the need for laborious fine-grain labeling.

We chose to focus on the foundational task of surgical step recognition – that is, parsing a procedure video into meaningful segments that represent the surgeon's workflow. Step recognition is a fundamental task for any avenue of surgical data science and it serves as a benchmark task in the field of surgical intelligence (Maier-Hein et al., 2017). While previous studies have explored step recognition for a single type of surgical procedure, laparoscopic Cholecystectomy (Bar et al., 2020; Twinanda et al., 2016), Cataract surgery (Yu et al., 2019; Zisimopoulos et al., 2018) and laparoscopic Sleeve Gastrectomy (Hashimoto et al., 2019), they did not assess whether their methods would perform well if applied to other types of surgeries and did not examine the ability to adapt to new types of procedures.

In the traditional apprenticeship model, known as the Halstedian method (Cameron, 1997), after witnessing one surgical procedure, a trainee should be capable of performing the procedure on her own and then teach it. In this study, we follow the analogous idea from the machine learning domain, where it is often referred to as transfer learning. In our case, the objective is to demonstrate that it is possible to first train and classify on a single surgical procedure, and then transfer knowledge to a different procedure type.

Transfer learning attempts to exploit a model that was pre-trained on one task, and apply its knowledge when training a different task, thus improving the overall generalization (Goodfellow et al., 2016). It is especially useful when the target task's dataset is relatively small, like in the surgical domain. Transfer learning has been proven to be a robust method in many ML challenges. Specifically in the computer vision domain, it enables achieving state-of-the-art results in object detection (Girshick et al., 2014; Girshick, 2015), image segmentation (Long et al., 2015; He et al., 2017), face identification (Taigman et al., 2015)

and video action recognition (Carreira and Zisserman, 2017). However, transfer learning tends to work better when the source task is related to the target task (Yosinski et al., 2014). In Figure 1, we demonstrate how such relation exists in the task of step recognition by comparing the view of *Adhesiolysis* in four different procedure types. In *Adhesiolysis*, the goal is to remove adhesions. In these procedures, the adhesions are abdominal, and their removal can be done with different tools. While the anatomy viewed changes between procedures, the action remains the same. Thus, we argue that adapting knowledge from one procedure to another is beneficial.

The standard approach for step recognition in previous studies is training two models for each procedure type. A deep ConvNet that extracts visual features and a time-series model that processes the features sequentially (Bar et al., 2020; Zisimopoulos et al., 2018; Hashimoto et al., 2019). The ConvNets are first trained with non-surgical datasets, e.g., ImageNet (Deng et al., 2009) and Kinetics-400 (Kay et al., 2017), but the obvious approach of transferring knowledge across different procedures has never been assessed before.

In this study, we suggest a new approach. We use a 3D ConvNet (Carreira and Zisserman, 2017; Wang et al., 2018), pre-trained for step recognition on Cholecystectomy (Bar et al., 2020). This model is used to extract feature representations from videos of three different laparoscopic procedures: Right Hemicolectomy, Sleeve Gastrectomy, and Appendectomy. We then explore various time-series architectures and focus on finding the best one for surgical domain adaptation. We also suggest a self-supervised initialization method that improves the performance of our time-series model. Finally, we compare our findings with the traditional approach described above.

## 2. Methods

Our overall approach involves (1) extracting feature representations from videos using a 3D ConvNet; and (2) training a time-series model on these features to predict a step label for each second of video.

### 2.1. Datasets

All datasets were randomly split into three subsets: training, validation, and test, with a ratio of 25% for the test and 20% of the remaining for the validation (Table 1). We provide a detailed description of the datasets and the steps workflow definition in Appendix A.

The annotation process is identical for all procedures types. Each video undergoes a rigorous annotation process by two different annotation specialists. The team of annotators underwent thorough training on labeling the workflow steps. Annotation process validity was confirmed in a previous study (Korndorffer Jr et al., 2020), in which an unbiased group of surgeons reviewed large portions of the Cholecystectomy cases and reported high agreement with our annotation method.

### 2.2. Time-series model architectures

We consider several time-series model architectures below and explore two main variants to process the temporal dimension: (1) 1D convolution layers and (2) recurrent layers. As the main contribution, we found a specific combination of the two that yields optimal

Table 1: Number of samples per subset for each of the target datasets.

|  | Total | Training | Validation | Test |
|---|---|---|---|---|
| Right Hemicolectomy | 205 | 123 | 31 | 51 |
| Sleeve Gastrectomy | 229 | 138 | 34 | 57 |
| Appendectomy | 852 | 511 | 128 | 213 |

performance. In what follows, we describe architectural details. In all cases, the final classification layer for all architectures is a fully connected layer, followed by a softmax function that predicts, for each second, a single surgical step.

**1D Convolution Layers (Conv1D).** As the short-term context is important when predicting a step for each second, we use a standard 1D convolution layer and apply it to the temporal dimension. In our experiments, we explore different kernel sizes in order to observe different temporal contexts (Han et al., 2020).

**Long Short-Term Memory (LSTM).** While Conv1D should be able to learn the context in a short temporal region of interest, it still lacks the larger scope of view and cannot link distant information. Hence, most recent studies use LSTM networks as their time-series model. We also explore the capabilities of LSTM to transfer knowledge and use a bidirectional LSTM in our experiments, thus not assuming any causality constraints.

**Time-Series Adaptation Network (TSAN).** Inspired by Ghosh and Kristensson (2017), our architecture fuses three Conv1D layers and two LSTM networks into a single architecture. The three Conv1D operate in parallel to a single bidirectional LSTM. The outputs of all four are concatenated and applied to an additional bidirectional LSTM, followed by a fully connected layer for classification (Figure 2).

**Sequence Sorting (SeSo).** Compared to other architectures, TSAN is a deeper network, with more parameters to train. The fact that the target surgical datasets are relatively small, especially compared to other video benchmarks (Kay et al., 2017), led us to explore a better initialization technique as an alternative to the random initialization.

We establish our initialization approach using an analogy of solving jigsaw puzzles (Noroozi and Favaro, 2016). In the temporal domain, this can be structured as correctly reassembling the shuffled segments of a video. We thus formulate a self-supervised training method as an initial task for step recognition.

More concretely, we split a video into nine segments and shuffle the order randomly. Then, we process each segment's feature vectors separately with a time-series model. We concatenate all nine segments' last layer output and feed the resulting representation to a classification head that predicts the correct order. We use SeSo to pre-train both TSAN and LSTM networks. We then remove the classification layer and finetune the networks on the step recognition task.

## 2.3. Implementation details

The training process of the 3D ConvNet follows the one of Bar et al. (2020). We use the exact same protocol of inflating a 2D image classification model, and convert it to a 3D model (Carreira and Zisserman, 2017). We first train the 3D model for video action

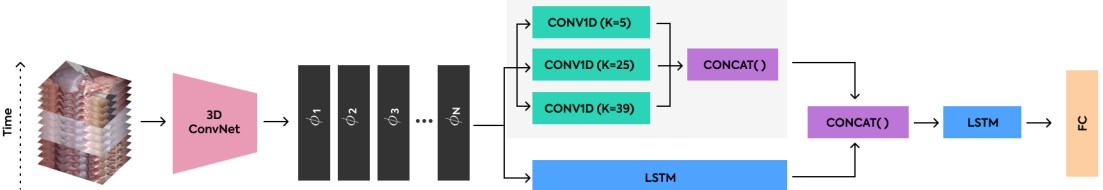

Figure 2: Time-Series Adaptation Network (TSAN) architecture. Combining three Conv1D layers with two LSTM, followed by a fully connected classification layer. $\phi_i$ indicates the feature representations extracted from the 3D ConvNet. $K$ denotes the kernel size of the 1D convolution layers.

recognition using the Kinetics-400 dataset (Kay et al., 2017) and then finetune the model on the surgical video dataset. The only difference compared to (Bar et al., 2020) is that instead of extracting the Softmax probability as the input for the temporal model we extract the bottleneck layer and use it as the features representation vectors. We consider each second as an independent sample and process a 64 frames clip around the target second. Stacking these vectors sequentially for each video yields a matrix of size $L \times N$. $L$ is the video's length (in seconds), and $N$ is the feature vector dimension ($N = 2049$ in all our experiments).

For the Conv1D, we explore three temporal kernel sizes, $K = 5, 25, 39$, and the output's length is matched to the input by padding, based on the kernel size. The output dimension of both the 1D convolution layer and the LSTM hidden layer is set to 128.

Each network architecture was trained for 100 epochs. We use SGD and set the learning rate to $10^{-3}$ for the Conv1D networks and $10^{-2}$ for the LSTM and TSAN networks. The loss function is the negative log-likelihood loss.

Since the features are extracted from the raw videos in advance, applying augmentations, like those used on images, is not feasible. Although it is possible to apply augmentations in advance and extract augmented features, on a practical level, it is computationally challenging. We did try exploring several known augmentations, but those showed no gain in performance. Thus, to apply some sort of data augmentation on our data and avoid overfitting, we apply two types of augmentations on the input features matrix. First, we used an out-of-body and non-relevant video segment detection by applying the method described by Zohar et al. (2020). We then mark each video second as either relevant or not, and randomly remove these seconds from the training with a probability of 0.5. We also use Dropout of 0.5 both on the input matrix and on the intermediate layers.

## 3. Results

### 3.1. Time-series model architecture experiments

We start by searching for an optimized architecture for surgical transfer learning. As a baseline, we use the traditional technique of training a 3D ConvNet on each target dataset, followed by training a bidirectional LSTM network on the resulting features. This is fully

Table 2: Comparing different time-series model architectures. The last column is a result of averaging the accuracy of all three target datasets: Right Hemicolectomy (RH), Sleeve Gastrectomy (SG), and Appendectomy (APPY). The first row is a result of using the standard approach without surgical transfer learning. Other table rows show the development of our suggested architecture. $K$ denotes the kernel size of the 1D convolution layers (C1D). $L$ denotes the number of LSTM layers.

| C1D (K=5) | C1D (K=25) | C1D (K=39) | LSTM (L=1) | LSTM (L=2) | With SeSo | Transfer learning | RH | SG | APPY | AVG |
|---|---|---|---|---|---|---|---|---|---|---|
| | | | | | | | | Accuracy | | |
| | | | ✓ | | | | 93.1 | 94.0 | 88.9 | 92.0 |
| ✓ | | | | | | ✓ | 89.2 | 85.3 | 80.0 | 84.8 |
| | ✓ | | | | | ✓ | 91.7 | 90.2 | 83.9 | 88.6 |
| | | ✓ | | | | ✓ | 92.1 | 90.7 | 84.9 | 89.2 |
| ✓ | ✓ | ✓ | | | | ✓ | 91.7 | 90.2 | 84.9 | 88.9 |
| | | | ✓ | | | ✓ | 91.7 | 92.3 | 88.8 | 90.9 |
| | | | ✓ | | ✓ | ✓ | 93.0 | 92.7 | 89.4 | 91.7 |
| | | | | ✓ | | ✓ | 93.3 | 91.7 | 89.8 | 91.6 |
| | | | | ✓ | ✓ | ✓ | 92.0 | 91.8 | 89.3 | 91.0 |
| ✓ | ✓ | ✓ | | ✓ | | ✓ | 92.13 | 93.9 | 90.0 | 92.0 |
| ✓ | ✓ | ✓ | | ✓ | ✓ | ✓ | **94.7** | **94.4** | **90.4** | **93.2** |

labeled training on each type of surgery – no surgical transfer learning is applied at this stage in the process.

We then evaluate several time-series models using Cholecystectomy features from a pre-trained 3D ConvNet. In Table 2, we evaluate various models described in Sec. 2.2. We report the test set accuracy by measuring the number of seconds in all test videos that are labeled correctly by each model.

At a high level, we see that TSAN, the combination of two LSTMs and three Conv1Ds, pre-trained with our self-supervised approach (Sec. 2.2), outperformed all other methods.

We also observe two other interesting results from the comparisons shown in Table 2. First, our transfer learning approach produces better results than using the traditional training method. And second, if one does apply transfer learning for the surgical domain, our method improves the results by about 2%, compared to a single LSTM network. The confusion matrix for each procedure type of TSAN with SeSo is provided in Appendix B.

### 3.2. Impact of the dataset size on model generalization

To further explore the generalization of our approach and its usability in the future for rapidly achieving high performance on smaller datasets, we study the impact of (labeled) training set size on the final accuracy results. We chose to focus on two variants that gave the best results for transfer learning, the LSTM network and our TSAN. Both are pre-trained using the SeSo method applied to features from Cholecystectomy.

To understand accuracy as a function of dataset size, we split the videos in the training sets of Right Hemicolectomy and Sleeve Gastrectomy into smaller subsets with $5, 10, 50$ and *all* training samples (*all* equals 123 and 138, respectively). For Appendectomy, as it is a

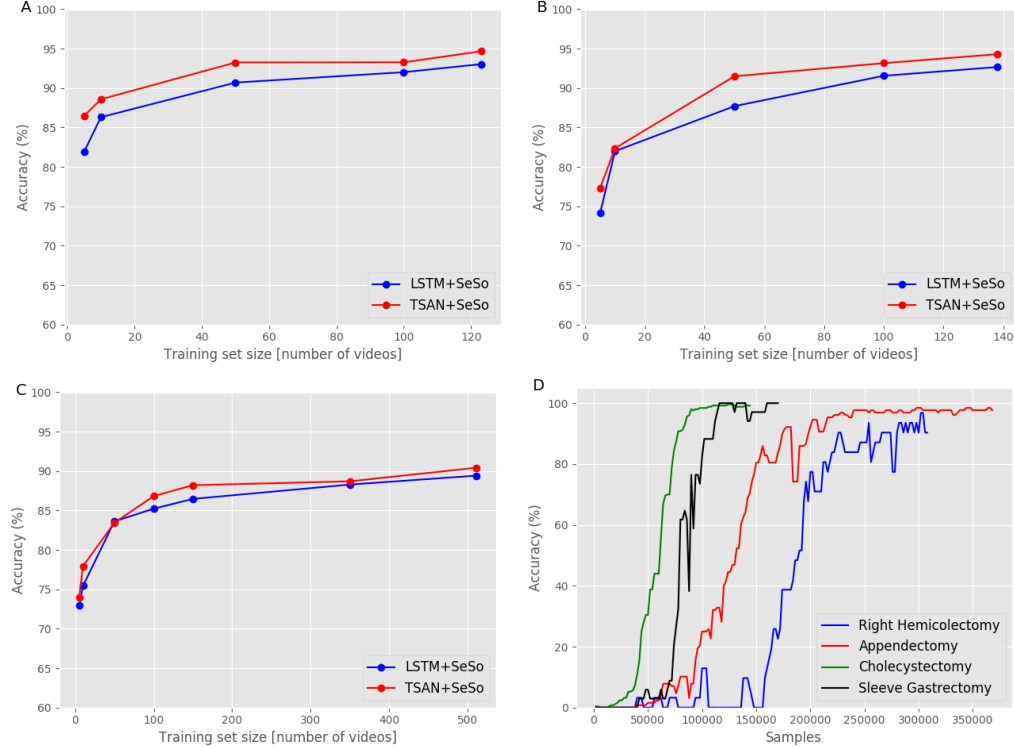

Figure 3: Evaluating the impact of the training set size on model generalization. (**A**) Right Hemicolectomy, (**B**) Sleeve Gastrectomy and (**C**) Appendectomy. We train the LSTM-SeSo and TSAN-SeSo, using smaller subsets of the original training set and measure the results on a fixed test set. (**D**) The validation accuracy curve when training the Sequence Sorting task *vs.* the number of samples used during training. The model trained using the Cholecystectomy data converges much faster compared to all other procedure types.

larger dataset, we added two additional subsets of 150 and 350 (*all* equals 511). The subsets are randomly selected and constructed so that each subset extends the previous smaller one. The test set is kept the same to enable a fair comparison.

In Figure 3, we show the accuracy values when training with different training set sizes. Our approach with SeSo generalized better for the majority of training set sizes. Furthermore, we see consistently high accuracy achieved between 100 and 200 videos.

### 3.3. Training the Sequence Sorting task

The SeSo initialization helps improve the results of the single LSTM and TSAN architectures. Especially for TSAN, this type of pre-training yields the best-performing architecture compared to other possibilities (Table 2).

To better understand the impact of SeSo on surgical transfer learning, we explore the effect of pre-training the time-series model on the source (Cholecystectomy) or the target datasets. We trained four TSAN models on the sorting task using all four datasets. Then, we fine-tuned the models on the step recognition task. We measure the accuracy on each

Table 3: Comparing step recognition accuracy results on the three target datasets after training the Sequence Sorting initialization task on either the source dataset (Cholecystectomy) or the target dataset.

| Step training dataset | SeSo training dataset | Accuracy |
|---|---|---|
| Right Hemicolectomy | Right Hemicolectomy | 94.5 |
| Right Hemicolectomy | Cholecystectomy | **94.7** |
| Sleeve Gastrectomy | Sleeve Gastrectomy | 94.2 |
| Sleeve Gastrectomy | Cholecystectomy | **94.3** |
| Appendectomy | Appendectomy | 89.9 |
| Appendectomy | Cholecystectomy | **90.4** |

of the three target datasets, first when pre-training using the source dataset (Cholecystectomy), and second when pre-training using the target dataset. Table 3 shows only a small improvement when using the source dataset to train the SeSo task. While this is surprising, it is likely due to the fact that the Cholecystectomy dataset is larger than the others. However, it supports the notion that self-supervised initialization can effectively exploit unlabeled data, even when it does not come from the target dataset.

In Figure 3.D, we plot the validation accuracy during the SeSo task training and demonstrate that the model also converges much faster on the Cholecystectomy dataset.

## 4. Conclusion

This work suggests a new approach to train surgical step recognition models by using surgical transfer learning. We show, for the first time, an analysis of transfer learning between different surgical procedures and our findings demonstrate that it is possible to transfer knowledge from one procedure to another, even when using relatively small target datasets. It is also the first study to explore surgical step recognition on Right Hemicolectomy and Appendectomy. To facilitate a robust domain adaptation, we explore various architectures and introduce a new time-series architecture, TSAN, optimized for model adaptation in the surgical domain. Moreover, we present a Sequence Sorting task as a pre-initialization method. The main advantage of this approach, besides that it improves TSAN performance when transferring knowledge from one surgery type to another, is the fact that it is trained with a self-supervised method.

Future work should explore how mutual learning of surgical step recognition, trained on several procedures simultaneously, will perform. Also, the ideas of domain adaptation presented in this study could be applied to other surgical related tasks, such as event detection, and it would be interesting to test our findings on such tasks.

Although significant progress has been made in recent years in the field of surgical intelligence, the next leap forward must focus on the practical implementation of applying artificial intelligence in the surgical domain. Solid evidence that these technologies can be generalized for various surgical procedures is essential for surgeons to embrace them as part of their daily routine, both inside and outside the operating rooms. We believe that surgical transfer learning and the ability to transfer knowledge between models in the surgical domain are key facilitators and will expedite the development of computer-assisted surgery in a wide range of surgical procedures. This study is a step in that direction.

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

## Appendix A. Detailed datasets description.

Table 4: A summary of our datasets characteristics. Describing the number of samples in each subset, the number of medical centers the data was curated from and the steps workflow categories definition.

| | Cholecystectomy | Right Hemicolectomy | Sleeve Gastrectomy | Appendectomy |
|---|---|---|---|---|
| Total | 1665 | 205 | 229 | 852 |
| Training set | 999 | 123 | 138 | 511 |
| Validation set | 250 | 31 | 34 | 128 |
| Test set | 416 | 51 | 57 | 213 |
| Medical centers | 12 | 4 | 2 | 3 |
| Step 1 | Preparation | Preparation | Preparation | Preparation |
| Step 2 | Adhesiolysis | Adhesiolysis | Adhesiolysis | Adhesiolysis |
| Step 3 | Dissection and Skeletonization | Mobilization and Dissection | Dissection of Greater Curvature | Dissection and Skeletonization |
| Step 4 | Division of Cystic Structures | Specimen Packaging | Gastric Transaction | Mobilization of Terminal Ileum or Cecum |
| Step 5 | Gallbladder Separation | Anastomosis | Reinforcement of Staple Line | Appendix Excision |
| Step 6 | Gallbladder Packaging | Specimen Retrieval | Specimen Extraction | Appendix Packaging |
| Step 7 | Final Inspection and Extraction | Final Inspection | Final Inspection | Final Inspection and Extraction |

The datasets characteristics are summarized in Table 4.

**Cholecystectomy.** This dataset contains 1665 videos curated from 12 different medical centers. Each second in the video was annotated, categorized into one of seven clinically-relevant surgical steps: (1) Preparation, (2) Adhesiolysis, (3) Dissection and Skeletonization, (4) Division of Cystic Structures, (5) Gallbladder Separation, (6) Gallbladder Packaging, and (7) Final Inspection and Extraction.

**Right Hemicolectomy.** This dataset contains 205 videos curated from four different medical centers. Each second in the video was annotated, categorized into one of seven clinically-relevant surgical steps: (1) Preparation, (2) Adhesiolysis, (3) Mobilization and Dissection, (4) Specimen Packaging, (5) Anastomosis, (6) Specimen Retrieval, and (7) Final Inspection.

**Sleeve Gastrectomy.** This dataset contains 229 videos curated from two medical centers. Each second in the video was annotated, categorized into one of seven clinically-relevant surgical steps: (1) Preparation, (2) Adhesiolysis, (3) Dissection of Greater Curvature, (4) Gastric Transaction, (5) Reinforcement of Staple Line, (6) Specimen Extraction, and (7) Final Inspection.

**Appendectomy.** This dataset contains 852 videos curated from three different medical centers. Each second in the video was annotated, categorized by one of seven clinically-

relevant surgical steps: (1) Preparation, (2) Adhesiolysis, (3) Dissection and Skeletoniza-tion, (4) Mobilization of Terminal Ileum or Cecum, (5) Appendix Excision, (6) Appendix Packaging, and (7) Final Inspection and Extraction.

## Appendix B. Confusion matrix analysis.

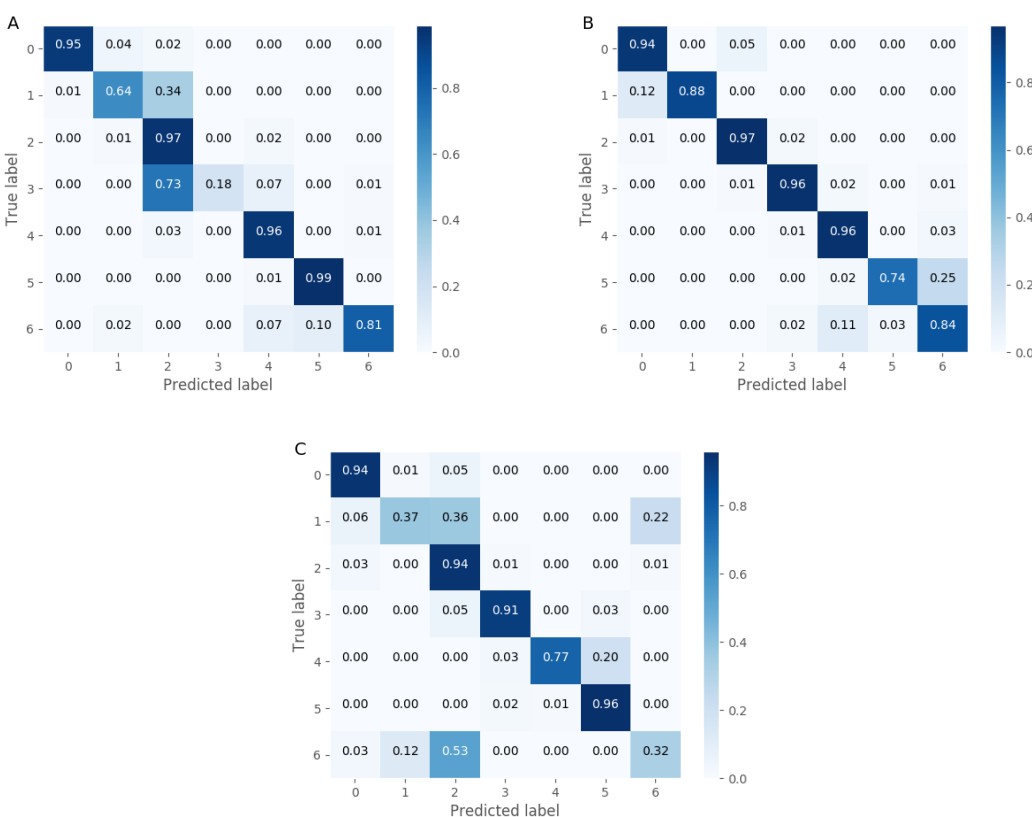

Figure 4: TSAN with SeSo confusion matrix on the test set for each procedure type. (A) Right Hemicolectomy, (B) Sleeve Gastrectomy and (C) Appendectomy.

