# OpenReview forum: "“Train one, Classify one, Teach one” - Cross-surgery transfer learning for surgical step recognition"
_MIDL.io/2021/Conference — MIDL 2021_

### Official Review · AnonReviewer1 · 2021-03-04

**Confidence:** 4
**Preliminary Rating:** 3
**Recommendation:** Poster
**Final Rating:** 4

**Summary:**

This work is an experimental study on several mechanisms to perform transfer learning in surgical phase recognition when starting from representations learned from a CNN pretrained on a Cholecystectomy operation. 1-D Cnns, LSTMs, and combinations of both are tested, as well as self-supervised pre-training. A couple of additional experiments on the impact of dataset sizes and of self-supervised pretraining are provided to complement the main performance comparison.

**Strengths:**

Although there is not much technical novelty, the paper is well-written, reads fast, and the experimental setup is convincing. I believe this work has practical interest, and the technical details are of enough clarity for someone to reproduce it. Experimental results look reasonable too.

**Weaknesses:**

I do not think there are major weaknesses in this paper, everything is clearly explained. I would point to the following as minor weaknesses:

1) The most important "issue" I see is the limited evaluation in terms of metrics of interest. I do not really think that accuracy alone is a good indicator of the real performance of a model in this context, it does not seem to tell the whole story. At the very least, it would have been interesting to see accuracy per-step, so that we can understand if maybe all models get common steps like "Preparation" right, but then some generalize better or worse to steps that are particular to a specific procedure, like gastric transaction. Also, isn't it possible to evaluate also how temporally consistent a model is on its predictions, by measuring the amount of wrong jumps it makes? I would rather have a model that produces a consistent workflow, it would be much easier to correct an entire segment that is wrong than seeing frames changing constantly, even if accuracy is 93.5% instead of 93.6%. All this could be super interesting info to add to the appendix, in my opinion.

2) For all the effort the authors put into making things reproducible, by reporting the tiniest detail, it is a pity that the datasets they use cannot be open-sourced to the community (can't they?)

**Deanonymize Review:**

no

**Detailed Comments:**

I don't think I have much to add to the above. Two minor aesthetic comments:

1) In Table 2, I don't see the necessity of the "Transfer Learning" column, as it is implicit by now that all starts from features extracted from a pretrained model.

2) Some typos are present, the authors may want to spell-check the paper. For example:
- we use a standard 1D convolution layer and apply them to the temporal dimension. -> apply it
- The three Conv1D operates in parallel -> operate

**Final Rating Justification:**

As I mentioned in my answer to the authors, the more thorough performance analysis reveals useful information regarding where the proposed approach is getting its improvement from. Since this seems to be from more complicated aspects of the procedure, I believe this increases the value of the method and therefore I review my recommendation to advice Strong Acceptance.

**Justification Of The Preliminary Rating:**

In my opinion, this paper could be accepted if there is enough room. It provides interesting results and enough technical details to be of practical usefulness for the community. I am not going for the Strong Accept because I find the technical novelty to be a bit limited.

**Paper Type:**

validation/application paper

**Questions To Address In The Rebuttal:**

I am happy with the paper as it is, but I would be a bit happier if the authors could do something to provide a more interesting evaluation considering other metrics of potential interest, as described in one of my previous comments.


**Special Issue:**

no

---

> ### Author Response · Authors · 2021-03-17
> **We thank the reviewer for reading the manuscript and providing us with these constructive remarks.**
>
> Addressing the weaknesses raised by the reviewer:
>
> *1) The most important "issue" I see is the limited evaluation in terms of metrics of interest. I do not really think that accuracy alone is a good indicator of the real performance of a model in this context, it does not seem to tell the whole story. At the very least, it would have been interesting to see accuracy per-step, so that we can understand if maybe all models get common steps like "Preparation" right, but then some generalize better or worse to steps that are particular to a specific procedure, like gastric transaction. Also, isn't it possible to evaluate also how temporally consistent a model is on its predictions, by measuring the amount of wrong jumps it makes? I would rather have a model that produces a consistent workflow, it would be much easier to correct an entire segment that is wrong than seeing frames changing constantly, even if accuracy is 93.5% instead of 93.6%. All this could be super interesting info to add to the appendix, in my opinion.*
>
> Answer: To address the metrics issues raised by the reviewer, we use our best model, TSAN + SeSo, and provide the reader with the test set confusion matrix for each procedure type. We added the confusion matrix to Appendix B. As the reviewer suggested, we also analyzed the accuracy values per-step for all model variants. This helped us to better understand the generalization of our approach for each step compared to other model architectures. The results that we got show significant improvement on steps with fewer samples compared to a minor to almost no improvement on more common steps. For example, in Right Hemicolectomy step 2 [mobilization and dissection] is very common while step 1 [Adhesiolysis] is rarer. Comparing the results of the following architectures: LSTM (L=1), LSTM (L=1) with Transfer learning, LSTM (L=1) + SeSo with Transfer learning, and our proposed architecture [TSAN + SeSo with Transfer learning] yield the following accuracy for step 1 and step 2:
>
> Right Hemicolectomy - step 1 adhesiolisis:  96.3, 94.1, 95.0, 96.6
>
> Right Hemicolectomy - step 2 mobilization and dissection: 41.7, 52.2, 56.9, 64.1
>
> Thus, on common steps the transfer learning approach, with or without SeSo, has a marginal impact, but on less common steps, which have low baseline accuracy, the impact of transfer learning is significant, especially for TSAN + SeSo.
>
> Regarding the temporally consistent metric, we do measure an additional metric that aims to evaluate step jittering, however, since accuracy is the common metric in previous studies and our focus on the temporal model exploration, adding an additional metric and explaining its development was not feasible under the paper’s space limitations.
>
> *2) For all the effort the authors put into making things reproducible, by reporting the tiniest detail, it is a pity that the datasets they use cannot be open-sourced to the community (can't they?)*
>
> Answer: The videos that support the findings of this study, unfortunately, contain restricted data that, due to IRB and licensing limitations, we are unable to make publicly available. However, as the reviewer mentioned, we tried to be as transparent as possible in terms of technical details to allow the reproducibility of the methods presented in this study.
>
> *3) In Table 2, I don't see the necessity of the "Transfer Learning" column, as it is implicit by now that all starts from features extracted from a pretrained model.*
>
> Answer: All the results shown in Table 2 start from features extracted from a pretrained model on Cholecystectomy except the first row which uses features of a model trained on each specific procedure type. The reason we add the "Transfer Learning" column is to enable the reader to compare, in a single table, between a baseline that uses the standard approach, without transfer learning, and our suggested method.
>
> *4) Some typos are present, the authors may want to spell-check the paper. For example:
> we use a standard 1D convolution layer and apply them to the temporal dimension. -> apply it
> The three Conv1D operates in parallel -> operate*
>
> Answer: Thank you for noticing. We fixed the typos.

---

### Official Review · ~Alberto_Gomez1 · 2021-03-05

**Confidence:** 4
**Preliminary Rating:** 3
**Recommendation:** Poster
**Final Rating:** 3

**Summary:**

The authors propose, and compare, a number of architectures for surgical step recognition, as well as different pre-training strategies to evaluate the performance on transfer learning between different types of surgeries. The paper is well written and well organised, and the main part of the methods are described in enough detail to allow for reproducibility.  The paper scope is appropriate and interesting for the MIDL community.


**Strengths:**

* First time transfer learning between surgical procedures has been proposed
* A novel architecture combining LSTM and multiple Conv1D layers that outperform other models
* Interesting analysis on the impact of the size of training data, a very scarce resource in medical imaging


**Weaknesses:**

* Between worst and best performing result there is barely a 4% difference; no indication of results statistics (standard deviations, quantiles, statistical significance) is provided so although in absolute terms the results seem good, relative to each other it is not clear how they compare; they see to all do similarly. The fact that there is no discussion about why, even the worst model performs so well, makes me wonder if a) the problem is just too easy to solve, would there be a simpler model that performs well? and b) what accuracy is required for the application of interest? how good is good enough?

* Authors should give more details on the 3D conv net. Moreover, why do they use a 3D conv net to extract features of each frame, instead of a 2D conv net? By using 3D convolutions (with the 3rd dimension being time), the 3DConvNet is already assimilating temporal information. This might be behind the little difference between the temporal network: temporal information is already learnt by the 3D conv net.

* The main motivation for this work is that "Regardless of the use case, such video-based applications must serve a wide variety of surgical procedures in order to be relevant, actionable, meet the surgeons’ clinical needs and provide them with value.". I don't find enough evidence to support this claim, because an application that serves just one type of surgery can also be relevant and provide value, if it is a challenging surgery type. This does not demerit the methodological work, but I think authors should better build the case for clinical need.

* Not a scientific point but something confusing about the paper: the title, abstract and intro include the "fancy phrase" of "see one, do one, teach one". Although I find it "fun" and even somewhat poetic, I don't really think that it fits the paper. I expected a paper on training with *one* sample, then transferring to *one* other model or something like that. When the phrase says "one", the reader needs to re-interpret as "not too many". This completely breaks the original underlying idea from the Halstedian method. So I would strongly suggest to remove this reference from title, abstract and introduction. Also because it adds no value to the paper, so it is an unnecessary and misleading distraction.

**Deanonymize Review:**

yes

**Detailed Comments:**

* 3D conv net details are missing: (n layers, kernels, activations, etc)

* "Each video’s features is a matrix of size L × N . Where L is the length of the video (in seconds)" so there is one video feature per second? why not one feature per frame? and what is the framerate?

* Fig 3: what are the units of the training set size? The number of videos? if so please add this to the caption or axis label.

**Final Rating Justification:**

Authors have actually answered some of my comments, and I thank them for that. Although some of my comments were not fully addressed I still think this is an interesting contribution for the MIDL community.

**Justification Of The Preliminary Rating:**

Overall the paper is interesting, so I think it may have a place at the conference. As I raised in the weakness section, the results and the feature extraction step are unclear and must be improved to make this paper a solid accept.

**Paper Type:**

methodological development

**Questions To Address In The Rebuttal:**

From the weaknesses I pointed out, I think authors should include a statistical significance test to see if the difference between the different architectures is actually significant; and should further justify the use of a 3Dconvnet (instead of a 2D conv net as feature extraction, or include comparison with using a 2D conv net.

**Special Issue:**

no

---

> ### Author Response · Authors · 2021-03-17
> **We thank the reviewer for the interest in our work and the helpful comments.**
>
> Below please find our responses to the issues raised:
>
> *1) Between worst and best performing result there is barely a 4% difference; no indication of results statistics (standard deviations, quantiles, statistical significance) is provided so although in absolute terms the results seem good, relative to each other it is not clear how they compare; they see to all do similarly. The fact that there is no discussion about why, even the worst model performs so well, makes me wonder if a) the problem is just too easy to solve, would there be a simpler model that performs well? and b) what accuracy is required for the application of interest? how good is good enough?*
>
> Answer: In order to address the problem raised by the reviewer we performed the following analysis: We reshuffled the dataset per procedure and created additional versions of the training, validation and test sets. We then trained our best model, TSAN + SeSo, on all versions and report the mean and standard deviation of the final accuracy. Here are the results:
>
> Right Hemicolectomy:
> [94.6, 94.0, 93.8, 93.8, 94.7],
> Mean = 94.18,
> STD = 0.392
>
> Sleeve Gastrectomy:
> [95.3, 95.3, 95.3, 94.4, 94.4],
> Mean = 94.94,
> STD = 0.441
>
> Appendectomy:
> [91.2, 90.5, 90.3, 89.7, 89.6],
> Mean = 90.26,
> STD = 0.582
>
> Given these results, a 4% gain is significant and provides more evidence to support our approach.
>
> Regarding the question what accuracy is required for the application of interest? how good is good enough?
>
> This is a good question and it depends on the application at hand. Offline analysis of surgical videos, e.g., for mapping purposes, can use high but not perfect accuracy, while a real-time system, working in the OR during the operation, would demand higher performance. We believe that applying methods, like the one suggested in this study, with the right dataset volume will lead to the point of high enough accuracy to serve a variety of surgical-related applications, inside and outside the OR.
>
> *2) Authors should give more details on the 3D conv net. Moreover, why do they use a 3D conv net to extract features of each frame, instead of a 2D conv net? By using 3D convolutions (with the 3rd dimension being time), the 3DConvNet is already assimilating temporal information. This might be behind the little difference between the temporal network: temporal information is already learnt by the 3D conv net.*
>
> Answer: To train the 3D ConvNet we follow the same training process as the one described in Bar et al. (2020). In the revised manuscript, we added additional information about the training process and described the only difference between our study and the one of Bar et al. (2020):
>
> **“The training process of the 3D ConvNet follows the one of Bar et al. (2020). We use the exact same protocol of inflating a 2D image classification model, and convert it to a 3D model (Carreira and Zisserman, 2017). We first train the 3D model for video action recognition using the Kinetics-400 dataset (Kay et al., 2017) and then finetune the model on the surgical video dataset. The only difference compared to (Bar et al., 2020) is that instead of extracting the Softmax probability as the input for the temporal model we extract the bottleneck layer and use it as the features representation vectors. We consider each second as an independent sample and process a 64 frames clip around the target second. Stacking these vectors sequentially for each video yields a matrix of size L×N. L is the video’s length (in seconds), and N is the feature vector dimension (N = 2049 in all our experiments).”**
>
> A more detailed description of how the 3D ConvNets are trained is provided via the Bar etl al. reference.
>
> We chose to use 3D ConvNets as it was shown to be a powerful architecture that yields superior performance on several video action recognition datasets (Carreira and Zisserman, 2017; Wang et al., 2018). As the reviewer remarked, 3D ConvNets learn some temporal information, however, to extract features for each second we use a window of 64 frames from a video encoded with 25 FPS. This yields a relatively short clip of 2.56 seconds around the target second, and thus the temporal context is short, especially compared to what the temporal models we explore in this paper need to handle, which is videos of tens of minutes. As all temporal models use the same set of features as input, they all gain from this short-term context learning and thus the comparison is fair.

---

> > ### Author Response · Authors · 2021-03-17
> > **Responses to the issues raised continue**
> >
> > *3) The main motivation for this work is that "Regardless of the use case, such video-based applications must serve a wide variety of surgical procedures in order to be relevant, actionable, meet the surgeons’ clinical needs and provide them with value.". I don't find enough evidence to support this claim, because an application that serves just one type of surgery can also be relevant and provide value, if it is a challenging surgery type. This does not demerit the methodological work, but I think authors should better build the case for clinical need.*
> >
> > Answer: We agree with the reviewer that even applications which serve a single procedure type are important and valuable in the clinical setting. However, we argue that when serving a variety of surgical procedures the value is significantly enhanced and thus strengthness the need for studying and applying AI-based systems on multiple procedure types.
> >
> > We made the following changes in that paragraph to make the statements clearer for the reader:
> >
> > **“Regardless of the use case there is value in creating video-based applications that are able to readily scale to serve a wide variety of surgical procedures. This will both broaden the impact of AI-based systems and also create more relevant, actionable tools that meet the needs of the broadest population of surgeons.”**
> >
> > *4) Not a scientific point but something confusing about the paper: the title, abstract and intro include the "fancy phrase" of "see one, do one, teach one". Although I find it "fun" and even somewhat poetic, I don't really think that it fits the paper. I expected a paper on training with one sample, then transferring to one other model or something like that. When the phrase says "one", the reader needs to re-interpret as "not too many". This completely breaks the original underlying idea from the Halstedian method. So I would strongly suggest to remove this reference from title, abstract and introduction. Also because it adds no value to the paper, so it is an unnecessary and misleading distraction.*
> >
> > Answer: The paraphrasing is done to emphasize what transfer learning, if utilized correctly, can do in the surgical intelligence domain. While the Halstedian method means see one surgical procedure, we referred to it as see one surgical procedure type. To make this a bit more clear for the reader we changed the relevant paragraph in the introduction to the following:
> >
> > **“In the traditional apprenticeship model, known as the Halstedian method (Cameron, 1997), after witnessing one surgical procedure, a trainee should be capable of performing the procedure on her own and then teach it. In this study, we follow the analogous idea from the machine learning domain, where it is often referred to as transfer learning. In our case, the objective is to demonstrate that it is possible to first train and classify on a single surgical procedure, and then transfer knowledge to a different procedure type.”**
> >
> > *5) 3D conv net details are missing: (n layers, kernels, activations, etc)*
> >
> > Answer: Please see our answer to remark 2 above.
> >
> > *6) "Each video’s features is a matrix of size L × N . Where L is the length of the video (in seconds)" so there is one video feature per second? why not one feature per frame? and what is the framerate?*
> >
> > Answer: We extract a feature vector for each second in each video. Thus, each video in the dataset is represented by a matrix, where L is the video length in seconds. The size of the vector per second is N. The videos are encoded with 25 FPS, and the input for the 3D ConvNet is a clip around the target second of 64 frames. This resolution of the temporal domain is more compact than having a feature vector per frame while still having the relevant temporal information. We add the following sentence in the Implementation details to make this a bit more clear:
> >
> > **“We consider each second as an independent sample and process a 64 frames clip around the target second. Stacking these vectors sequentially for each video yields a matrix of size L×N. L is the video’s length (in seconds), and N is the feature vector dimension (N = 2049 in all our experiments).”**
> >
> > *7) Fig 3: what are the units of the training set size? The number of videos? if so please add this to the caption or axis label.*
> >
> > Answer: We added “number of videos” to the axis label in the revised manuscript.

---

> ### Author Response · Authors · 2021-03-22
> **Unfortunately, we accidentally marked all our responses with visibility of "program chairs and authors" only.**
>
> We changed the visibility settings to fix this now.

---

> ### Author Response · Authors · 2021-03-23
> **Unfortunately, we accidentally marked all our responses with visibility of "program chairs and authors" only.**
>
> We changed the visibility settings to fix this now.

---

### Official Review · AnonReviewer3 · 2021-03-08

**Confidence:** 3
**Preliminary Rating:** 2

**Summary:**

A deep-learning-based approach for surgical step recognition is described. It consists of two components - a feature extractor (3D ConvNet) and a time-series model (Conv1D / LSTM/ TSAN). Video data of four different laparoscopic surgeries is used and combined by transfer learning. Furthermore, a self-supervised learning strategy by temporal sequence sorting is introduced.

**Strengths:**

- The general idea of combining data of different laparoscopic surgeries due to same / similar steps is reasonable. However, the applied strategy of freezing a feature extractor trained on a single surgery is not clear to me. Please motivate.
- The idea of unsupervised pre-training via sequence sorting is neat and easy to implement.

**Weaknesses:**

- Please describe your data in more detail. What is the size and dimension of your data? I assume one temporal axis, 2 spatial ones and one channel axis (RGB)? How is the temporal resolution? Please also describe the steps of the Cholecystectomy cases. I would prefer a data section in front of the methods section.
- The paper is quite hard to follow and should be restructured. As mentioned before, describe your data first and also detail the 3D ConvNet training.
- The motivation part within the introduction is weak. Why do we need surgical step recognition?


**Deanonymize Review:**

no

**Detailed Comments:**

- I suggest to plot a confusion matrix of the best learning setup, to give a better impression to the reader which steps are easy to recognize and which tend to be misclassified. Furthermore, one can directly see the number of classes (steps) and for each class the corresponding number of video frames.
- Figure 3D does not belong to Figure 3A-C which is confusing for the reader.
- You state "And second, if one does apply transfer learning for the surgical domain,our method improves the results by about 2%, compared to a single LSTM network." However, if I compare a single LSTM without transfer learning and with transfer learning, I can see a decrease of 1.1% (90.2% - 90.9%) in the averaged accuracy. Results are quite mixed. I am not sure wheter your baseline experiment (using surgery-specific feature extractors) is sensible. Why don't you train the networks on all data sets in common? Or at least the feature extractor? How are the baseline results using SeSo?

**Justification Of The Preliminary Rating:**

Due to several issues and open questions (mentioned above), I do not recommend the paper for acceptance in its current form. I am willing to change my rating to weak accept if identified weaknesses are addressed in the rebuttal period.

**Paper Type:**

both

**Questions To Address In The Rebuttal:**

- How is the 3D ConvNet trained? Please describe in a few sentences (Target, Loss,...). The 3D ConvNet processes each time frame individually, right?
- You state "Since the features are extracted from the raw videos in advance, applying augmentations,like those used on images, is not feasible". Why isn't it possible to perform offline data augmentation, meaning augmenting the raw videos and afterwards applying the 3D ConvNet for feature extrection on the augmented data.
- Why do you use a bidirectional LSTM? But within the sequence sorting you demand temporal causality.


**Special Issue:**

no

---

> ### Author Response · Authors · 2021-03-17
> **We would like to thank the reviewer for the valuable comments on the manuscript. We appreciate the opportunity to clarify our research and have made a few changes in the manuscript based on the proposed suggestions.**
>
> *1) The general idea of combining data of different laparoscopic surgeries due to same / similar steps is reasonable. However, the applied strategy of freezing a feature extractor trained on a single surgery is not clear to me. Please motivate.*
>
> Answer: On the practical level, commonly used frameworks for step recognition use two-step processing: (1) a deep ConvNet and (2) a temporal model. The ConvNet is the more demanding model, in terms of compute resources, training run time, and number of labels. Our motivation in this work is to offer a new approach in which the ConvNet needs to only be trained once, and from that point, it is possible to extract features on new procedure types and only train the temporal model. This allows more practical scaling to multiple procedure types with relatively small datasets.
>
> From a theoretical perspective, it has been shown that using features extracted with deep ConvNets trained on large datasets, such as ImageNet, produces features that can be used for finetuning on other tasks and yield high performance. Inspired by these previous studies our goal was to assess how a similar approach would work on the challenging surgical domain.
>
> *2) Please describe your data in more detail. What is the size and dimension of your data? I assume one temporal axis, 2 spatial ones and one channel axis (RGB)? How is the temporal resolution? Please also describe the steps of the Cholecystectomy cases. I would prefer a data section in front of the methods section.*
>
> Answer: The data dimension for the 3D ConvNets is Temporal (T=64), Channel (C=3), Height (H=224), and Width (W=224). The videos are encoded using 25 FPS. Since the paper focuses on temporal model exploration, and given space limitations, we chose to describe in more detail the information regarding the dimension of the input for the temporal models, assuming the feature extraction models are already trained. Nonetheless, we have added more information about the 3D ConvNet training (see next answer) and point the reader to Bar et al. (2020) reference, which describes in greater detail the data dimensions of the 3D ConvNets.
>
> We have modified the paper by adding a detailed description of the Cholecystectomy dataset to the table in Appendix A and moved the Datasets subsection to the beginning of the Methods section.
>
> *3) The paper is quite hard to follow and should be restructured. As mentioned before, describe your data first and also detail the 3D ConvNet training.*
>
> Answer: As the reviewer suggests, we have moved the Datasets subsection to the beginning of the Methods section. Regarding the 3D ConvNet training, we follow the same training process as the one of Bar et al. (2020). We added additional information, at the beginning of the Implementation details section, about the training process and state the only difference between our study and the one of Bar et al. (2020):
>
> **“The training process of the 3D ConvNet follows the one of Bar et al. (2020). We use the exact same protocol of inflating a 2D image classification model, and convert it to a 3D model (Carreira and Zisserman, 2017). We first train the 3D model for video action recognition using the Kinetics-400 dataset (Kay et al., 2017) and then finetune the model on the surgical video dataset. The only difference compared to (Bar et al., 2020) is that instead of extracting the Softmax probability as the input for the temporal model we extract the bottleneck layer and use it as the features representation vectors. We consider each second as an independent sample and process a 64 frames clip around the target second. Stacking these vectors sequentially for each video yields a matrix of size L×N. L is the video’s length (in seconds), and N is the feature vector dimension (N = 2049 in all our experiments).”**
>
> A more detailed description of how the 3D ConvNets are trained is provided via the Bar etl al. reference.
>
> *4) The motivation part within the introduction is weak. Why do we need surgical step recognition?*
>
> Answer: We made the following modification in the revised version to better explain the motivation behind the need for surgical step recognition:
>
> “We chose to focus on the foundational task of surgical step recognition – that is, parsing a procedure video into meaningful segments that represent the surgeon’s workflow. **Step recognition is a fundamental task for any avenue of surgical data science and it serves as a benchmark task in the field of surgical intelligence (Maier-Hein et al., 2017).”**

---

> > ### Author Response · Authors · 2021-03-17
> > **Responses to the issues raised continue (1)**
> >
> > *5) I suggest to plot a confusion matrix of the best learning setup, to give a better impression to the reader which steps are easy to recognize and which tend to be misclassified. Furthermore, one can directly see the number of classes (steps) and for each class the corresponding number of video frames.*
> >
> > Answer: As suggested, we have added the normalized confusion matrix for each procedure type in the test set (in Appendix B).
> >
> > *6) Figure 3D does not belong to Figure 3A-C which is confusing for the reader.*
> >
> > Answer: Our goal was to have all Results related figures in the same place. We agree that Figure 3D shows different experiment results than 3A-C; unfortunately, due to space limitations, we were not able to split the figure apart.
> >
> > *7) You state "And second, if one does apply transfer learning for the surgical domain,our method improves the results by about 2%, compared to a single LSTM network." However, if I compare a single LSTM without transfer learning and with transfer learning, I can see a decrease of 1.1% (92.0% - 90.9%) in the averaged accuracy. Results are quite mixed. I am not sure wheter your baseline experiment (using surgery-specific feature extractors) is sensible. Why don't you train the networks on all data sets in common? Or at least the feature extractor? How are the baseline results using SeSo?*
> >
> > Answer: As the reviewer comments, applying transfer learning on a single LSTM provides lower performance. This is one of the reasons to search for a better architecture for surgical transfer learning. The comparison that we are most interested in is a single LSTM without transfer learning (the standard approach done in previous studies) vs. TSAN + SeSo with transfer learning. This comparison shows an improvement in results when using our suggested method.
> >
> > We set the single LSTM without transfer learning as the baseline to compare to since this is the common method suggested in recent studies.
> >
> > Training on all datasets simultaneously is a very interesting concept. In the Conclusion section as part of the future work paragraph we wrote the following:
> >
> > **“Future work should explore how mutual learning of surgical step recognition, trained on several procedures simultaneously, will perform.”**
> >
> > Indeed training step recognition in a multi-task manner makes a lot of sense and this is something that we are working on now.
> >
> > Regarding the comment of the baseline results using SeSo, in Table 3 we showed that it is more beneficial to use the largest (Cholecystectomy) dataset for both the features and the SeSo initialization. We did try to train the SeSo when using the features of the target datasets, without any transfer learning, but since the datasets are small compared to Cholecystectomy the SeSo task was not stable to train and did not converge well. This is in fact one of the main practical takeaways of our study: that given a single large-scale dataset, one can train a single 3D ConvNet for feature extraction and a single matching SeSo model. With this in hand, it is fast and easy, in terms of compute limitation and data demands, to transfer to new procedures, which makes the framework highly practical and generic.
> >
> > *8) How is the 3D ConvNet trained? Please describe in a few sentences (Target, Loss,...). The 3D ConvNet processes each time frame individually, right?*
> >
> > Answer: The 3D ConvNet considers each video’s second as an independent sample, and extracts the features for that second based on a clip of 64 frames surrounding it. As mentioned in our answer to comment 3 above, we added more information in the Implementation details section about the training of 3D models, however, due to space limits, describing other aspects such as the loss function, optimization process, and other hyperparameters is not feasible, and we point the reader to the relevant reference of Bar et al. (2020) which provided detailed information of how these models are trained.

---

> > > ### Author Response · Authors · 2021-03-17
> > > **Responses to the issues raised continue (2)**
> > >
> > > *9) You state "Since the features are extracted from the raw videos in advance, applying augmentations,like those used on images, is not feasible". Why isn't it possible to perform offline data augmentation, meaning augmenting the raw videos and afterwards applying the 3D ConvNet for feature extrection on the augmented data.*
> > >
> > > Answer: In this study, we chose to focus on exploring temporal models, which use a given set of fixed features matrix per video and not on the feature extraction model. The reviewer is right and one can apply augmentations on the clip level and thus extract augmented features. In fact, we have done this in the early stages of this study, and the impact on the results was not significant. On a practical level, it is also extremely expensive, in terms of storage and computation, to extract and store many data variants, especially when no gain was shown in our preliminary evaluation. The paragraph we had in the paper states that given the fixed set of features, applying augmentations is not intuitive as applying augmentations on images and videos. We modified the related paragraph to explain our statement better:
> > >
> > > “Since the features are extracted from the raw videos in advance, applying augmentations, like those used on images, is not feasible. **Although it is possible to apply augmentations in advance and extract augmented features, on a practical level, it is computationally challenging. We did try exploring several known augmentations, but those showed no gain in performance.** Thus, to apply some sort of data augmentation on our data and avoid overfitting, we apply two types of augmentations on the input features matrix.”
> > >
> > > *10) Why do you use a bidirectional LSTM? But within the sequence sorting you demand temporal causality.*
> > >
> > > Answer: The Sequence Sorting task acts as a warm initialization of the temporal models. When training the sorting task we still use the same network architecture, including the bidirectional LSTM. The way we implemented the SeSo training is by splitting the video into nine segments, and the model objective is, given a mixed order of segments, to detect the correct order. The input to the temporal model is the sequentially stacked features of each one of the nine segments, and the output is the hidden layer of the last second in that segment. After processing all nine segments we concatenate the resulting representation of the last hidden layer from all segments and use another linear layer to classify the correct order. Thus, we don’t demand temporal causality when processing each segment independently and the model learns the right temporal order of the seconds by processing start-to-end and end-to-start, simultaneously.

---

> ### Author Response · Authors · 2021-03-22
> **Unfortunately, we accidentally marked all our responses with visibility of "program chairs and authors" only.**
>
> We changed the visibility settings to fix this now.

---

> ### Author Response · Authors · 2021-03-23
> **Unfortunately, we accidentally marked all our responses with visibility of "program chairs and authors" only.**
>
> We changed the visibility settings to fix this now.

---

### Meta-Review · Area_Chair1 · 2021-03-29

**Recommendation:** Accept (Poster)

**Metareview:**

The reviewer agree that the paper shows an rounded analysis of an intersting problem, albeit with little methodological novelty. The authors hae taken onboard the reviews and improved their manuscript accordingly.

**Paper Type:**

validation/application paper

---

### Decision · Program_Chairs · 2021-03-31

Accept